# Epipolar Geometry Improves Video Generation Models

## Abstract

Video generation models have progressed tremendously through large latent diffusion transformers trained with rectified flow techniques. Yet, despite these advances, these models still struggle with geometric inconsistencies, unstable motion, and visual artifacts that break the illusion of realistic 3D scenes. 3D-consistent video generation could significantly impact numerous downstream applications in generation and reconstruction tasks. This work explores how simple epipolar geometry constraints can improve modern video diffusion models trained on internet-scale datasets. Despite their massive training data, these models often fail to capture the fundamental geometric principles underlying all visual content. While traditional computer vision methods are often non-differentiable and computationally expensive, they provide reliable, mathematically grounded signals for 3D consistency evaluation. We demonstrate that aligning diffusion models through a preference-based optimization framework using pairwise epipolar geometry constraints yields videos with superior visual quality, enhanced 3D consistency, and significantly improved motion stability. Our approach offers an efficient alignment strategy that enforces established geometric principles without requiring end-to-end differentiability. Evaluation shows that our method outperforms baseline models and alternative alignment approaches across various metrics. By bridging the gap between data-driven deep learning and classical geometric computer vision, we present a practical method for generating more spatially consistent videos without compromising visual quality or requiring explicit 3D supervision.

## 1 Introduction

Video generation has witnessed remarkable progress in recent years, with newer models [1–6] producing increasingly realistic content from text and image conditions. This rapid advancement has spurred researchers to repurpose these powerful video models for broader applications, including animation [7], virtual worlds generation [8], and novel view synthesis [9].

Video diffusion models are trained on vast volumes of data, encoding rich priors about the visual world and its dynamics. Through extensive training, these models develop a strong understanding of object appearance, motion patterns, and scene composition. As a result, many recent works aim to utilize the priors from latent video diffusion models in various downstream tasks [10–12]. Despite this remarkable progress, these models still struggle to maintain perfect 3D consistency throughout generated sequences. Current video models often produce content with geometric inconsistencies, unstable motion, and perspective flaws, even though almost all training data is 3D consistent. Some approaches for enhancing 3D consistency rely on noise optimization [13], explicit guidance through point clouds [14, 15], or camera parameters [16]. Nevertheless, inaccurate control signals can constrain the model's generative capabilities, and the latent space optimization typical in diffusion training makes it difficult to compute direct geometric losses on the final outputs.

Submitted to 39th Conference on Neural Information Processing Systems (NeurIPS 2025). Do not distribute.

With the rising popularity of reinforcement learning for model alignment [17–19], post-training alignment has recently gained more attention in diffusion model research as an alternative approach to improve model capabilities. Methods such as VideoReward [20] have finetuned vision-language models on a large-scale human preference data, enabling direct supervision through the reward model. However, it relies on human-annotated motion quality scores (1 to 5), which can introduce noisy signals into the training process and are expensive to collect. Human judgments about video quality are inherently subjective and may not consistently capture the geometric principles that ensure proper 3D consistency. The gap between subjective human evaluations and objective geometric requirements creates an opportunity for alignment methods that leverage more mathematically grounded metrics for video quality assessment.

We propose a simple approach that bridges modern video diffusion models with classical computer vision algorithms. Rather than incorporating explicit 3D guidance during generation, we use well-established non-differentiable geometric constraints as reward signals in a preference-based finetuning framework. Specifically, we leverage an epipolar geometry constraint: assessing 3D consistency between frames. By sampling multiple videos conditioned on the same prompt, we generate diverse camera trajectories that typically vary in geometric coherence. The quality of these trajectories is well-captured by epipolar geometry metrics, providing a reliable signal for identifying which generations better adhere to projective geometry principles. This insight enables us to rank videos based on their adherence to epipolar constraints, creating training pairs that guide the model toward improved geometric consistency.

Our method implements this through Direct Preference Optimization (DPO) [17], requiring only relative rankings rather than absolute reward values. This approach bypasses the difficulties of directly using non-differentiable computer vision algorithms in the training loop. DPO only needs to determine which output better adheres to the geometric constraints. By finetuning the model to prioritize generations that satisfy these classical geometric constraints, we guide it towards generating inherently more 3D-consistent videos, without restricting its creative capabilities or requiring explicit 3D supervision. As shown in Figure 1, this results in enhanced 3D consistency, smoother camera trajectories, and fewer artifacts compared to the baseline model.

While simple in nature, this paper shows that a basic geometric constraint, described in 1982 [21], can recover what video models fail to do, even after large-scale training on billion-scale data: 3D consistency. In summary, the key contributions are as follows:

**Epipolar Geometry Optimization:** We introduce a method for finetuning video diffusion models using epipolar geometry constraints as reward signals, particularly leveraging the Sampson distance to enhance 3D video consistency without needing differentiability. The models finetuned with the simple yet reliable signal from classical computer vision algorithms achieve superior consistency and quality, significantly reducing artifacts and unstable motion trajectories in generated content. Our approach demonstrates that aligning models with fundamental geometric principles leads to visually superior results while preserving the model's ability to generate diverse and creative content.

**Comprehensive Evaluation Framework:** We develop an extensive evaluation protocol that measures both perceptual quality and 3D consistency and adherence to projective geometry principles across diverse generation scenarios. We evaluate text and image-to-video finetuned models, exploring the impact of geometry-aware finetuning on a large set of metrics.

**Large-Scale Preference Dataset:** We create and release a large dataset of over 162,000 generated videos annotated with 3D scene consistency metrics, enabling further research in geometry-aware video generation. This dataset includes diverse prompts spanning natural landscapes, architectural scenes, and dynamic environments, each with multiple video generations.

## 2   Related Work

We structure the related work section into generative models and post-training methods to adapt them.

### 2.1   Video Generation Models

Recent advances in video generation have been dominated by closed-source models developed by well-resourced technology companies. These models, trained on large proprietary datasets with

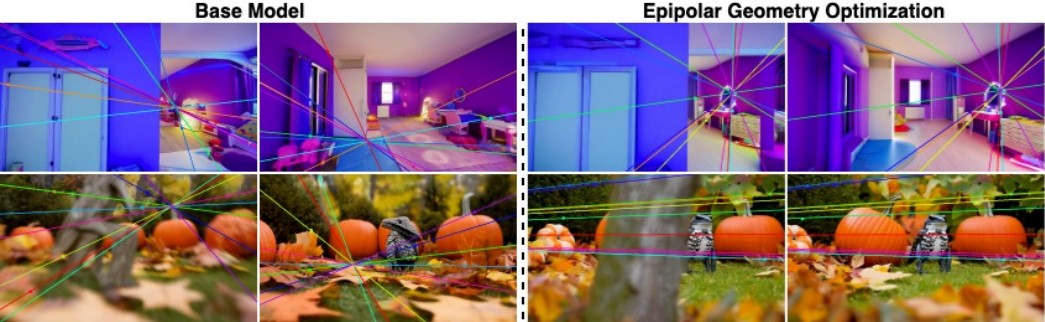

Figure 1: First and middle frame from videos generated by baseline and our epipolar-aligned model. The baseline model produces geometrically inconsistent outputs with artifacts and unnatural motion trajectories visible in distorted structures. Our model, finetuned with epipolar error, generates visibly improved results with smoother camera trajectories, reduced artifacts, and enhanced 3D consistency.

computational resources beyond academic reach, have demonstrated remarkable capabilities while revealing limited architectural details. Notable releases include OpenAI's Sora [1], which marked a significant leap in long-form video synthesis; Runway's Gen-2 and Gen-3 [22]; Luma AI's video models [23]; Pika Labs models [6]; and Google DeepMind's Veo series [2]. While these systems produce impressive results, their closed nature limits opportunities to finetune them or apply them to other vision tasks.

More recently, open-source large latent diffusion models have become available, increasing interest in improving video generators. Stable Video Diffusion [24] developed efficient training strategies for latent video diffusion. Hunyan-Video [5] presented a systematic approach to scaling models, LTX-Video [25] introduced optimizations for real-time generation, and Wan-2.1 [4] introduced an efficient 3D Variational Autoencoder [26] with expanded training pipelines. Wan-2.1 offers models for text-to-image and video-to-image in 1.3B and 14B parameter versions, enabling researchers to explore adaptation techniques for various downstream tasks.

These video diffusion models are trained on enormous data volumes covering more content variety than specific applications need, making domain-aware alignment valuable for specialized tasks. Geometry-aware finetuning allows general-purpose models to maintain creative flexibility while ensuring adherence to physical principles like 3D consistency. V3D [12] finetunes models to generate 360 orbit frames for 3D reconstruction, while VideoReward [20] introduced a framework for reinforcement learning-based video model alignment. However, prior methods rely on subjective human preferences or vision language models [27] trained to mimic them. In contrast, our approach optimizes against mathematical rules from epipolar geometry, providing a clean signal that aligns models with fundamental 3D consistency principles rather than subjective judgments.

## 2.2 Diffusion Models Alignment

Since image and video latent diffusion models are trained on internet-scale noisy data, efficient finetuning, and alignment strategies have emerged as an active research area. Latent image diffusion models [28, 29] finetune models on data highly ranked by the aesthetics classifier [30]. DRAFT [31] and AlignProp [32] further explore this paradigm by tuning the diffusion model to maximize the reward function directly. DPOK [33] and DDPO [34] expand the paradigm to introduce distributional constraints. Diffusion-DPO [35] introduces the Direct Preference Optimization algorithm into diffusion model alignment. In contrast to other approaches, DPO does not require direct access to the reward model and can be trained with only pairwise preference data. Additionally, this eliminates the need to decode the final denoised sample, which can be finetuned directly in latent space, significantly improving training efficiency. Recently, VideoReward [20] adapted Diffusion-DPO for video alignment, effectively aligning video generation with human preferences. Yet, all these approaches focus on optimizing for subjective and noisy human evaluation. Lately, DSO [36] employs DPO to align 3D generators with physical soundness, and PISA explicitly [37] improves the physical stability of video generators with a multi-component reward function. Our method leverages classical computer vision algorithms to provide objective, mathematically grounded preference signals based

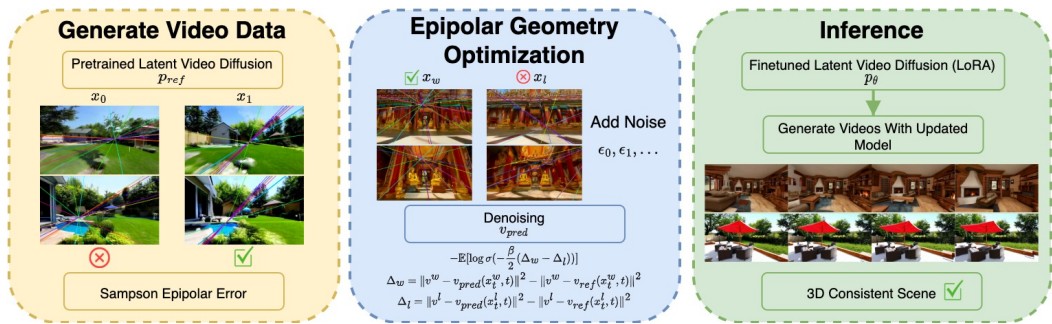

Figure 2: **Epipolar Geometry Optimization pipeline.** Our approach: (1) Generate diverse videos using pretrained generators [4] and leverage the Sampson epipolar error to identify 3D consistent vs. inconsistent samples; (2) Train policy $p_\theta$ using Flow-DPO [20] to prefer geometrically consistent outputs; (3) Apply the updated policy to enhance 3D consistency in the base video diffusion model.

on epipolar geometry, resulting in more reliable and consistent alignment with 3D physical principles. However, creating an explicit, robust, differentiable geometry reward model is challenging due to the complexity of accurately modeling and evaluating 3D consistency across diverse scenes. Our method addresses these challenges by leveraging classical computer vision algorithms to provide objective, physically grounded preferences based on epipolar geometry, resulting in consistent alignment.

## 3 Method

We aim to align pretrained video diffusion models to generate geometrically consistent 3D scenes from text or image prompts. To address this, we propose an alignment strategy that leverages classical epipolar geometry constraints within a preference-based optimization framework. Traditional reinforcement learning approaches [18, 38] require explicit reward functions and access to final samples which is impractical for video models due to the absence of robust differentiable reward models and the prohibitive computational cost of the denoising process. Our key observation is that while classical epipolar geometry constraints do not produce a smooth, globally comparable loss surface across different scene types (e.g., indoor vs. outdoor scenes may exhibit different absolute error magnitudes due to variations in matchable feature counts), the relative intra-prompt error measurements remain consistent. When generating multiple video sequences with fixed conditioning, the stochastic nature of diffusion sampling produces outputs with varying degrees of geometric consistency. Epipolar error metrics are an effective tool to quantify relative 3D consistency, with higher values reliably indicating lower geometric consistency. This finding aligns with the direct preference optimization (DPO) paradigm, which requires only a relative metric to determine preference between output pairs rather than absolute reward values. The pairwise comparison nature of DPO eliminates the need for a globally normalized reward function, instead leveraging the reliable local ranking provided by epipolar geometry measurements to guide model alignment toward more geometrically consistent video generation.

### 3.1 Objective Function

Given the pretrained video generator $p_{\text{ref}}$ that takes a text prompt and an optional first frame conditioning $I$ and generates video samples $x_0 \sim p_{\text{ref}}(x_0|T, I^*)$, where $I^* \in \{I, \emptyset\}$ we want to learn the model $p_\theta$ which is optimized to generate 3D-consistent video sequences. The one choice would be to optimize it with the following objective:

$$\max_\theta \mathbb{E}_{(T,I^* \in \{I,\emptyset\}) \sim \mathcal{D}_c, x_0 \sim p_\theta(x_0|T,I^*)} [r(x_0)]$$
$$- \beta \mathbb{D}_{\text{KL}} [p_\theta(x_0|T, I^*) \| p_{\text{ref}}(x_0|T, I^*)], \qquad (1)$$

where $r(x_0)$ is a reward function that outputs 3D consistency scores, $\mathcal{D}_c$ are samples from the reference model. The reward is maximized while the optimized model $p_\theta$ is kept close to the reference model $p_{\text{ref}}$ via a KL-divergence term weighted by the hyperparameter $\beta$. This formulation directly encourages the model to generate videos with improved geometric consistency. However,

this formulation presents a few critical practical challenges. First, the reward function $r(x_0)$ relies on classical computer vision algorithms that are non-differentiable, making direct gradient-based optimization infeasible. Second, evaluating this reward function requires complete video generation and subsequent geometric analysis, which is highly time-consuming for training large video diffusion models. These constraints make traditional reinforcement learning approaches impractical for our setting, and motivate our adoption of Direct Preference Optimization (DPO) [17, 35], which was originally designed for scenarios where direct reward optimization is similarly challenging.

Assuming a fixed dataset of $\mathcal{D}(\{x, x_0^w, x_0^l\})$ which consists of condition $c$ (text, image), and a pair of samples from the $p_{\mathrm{ref}}$ such that $x_0^w$ has higher reward value than $x_0^l$ ($x_0^w \succ x_0^l$).

Diffusion-DPO [35] aligns diffusion models with human preferences by directly solving eq. (1) analytically. It interprets alignment as a classification problem and optimizes a policy to satisfy the preferences through supervised training. The Diffusion-DPO objective $\mathcal{L}_{\mathrm{DD}}(\theta)$ is given by:

$$-\mathbb{E}\Bigg[ \log \sigma \Bigg( -\frac{\beta}{2} \Big( \|\epsilon^w - \epsilon_\theta(\mathbf{x}_t^w, t)\|^2 - \|\epsilon^w - \epsilon_{\mathrm{ref}}(\mathbf{x}_t^w, t)\|^2$$
$$- \big( \|\epsilon^l - \epsilon_\theta(\mathbf{x}_t^l, t)\|^2 - \|\epsilon^l - \epsilon_{\mathrm{ref}}(\mathbf{x}_t^l, t)\|^2 \big) \Big) \Bigg) \Bigg], \tag{2}$$

where $x_t^* = (1-t)\, x_0^* + t\, \epsilon^*, \epsilon^* \sim \mathcal{N}(0, \mathbf{I})$. The superscript $^* \in \{w, l\}$ denotes either $w$ for the sample with a higher score or $l$ for a sample with a lower score, $\epsilon^*$ is a ground truth or predicted noise by a diffusion model. The expectation is taken over samples $\{\mathbf{x}_0^w, \mathbf{x}_0^l\} \sim \mathcal{D}$ and the noise schedule $t$.

For rectified flow models [39–41] the noise vector $\epsilon^*$ is related to the velocity field $v^*$ following [20]:

$$\|\epsilon^* - \epsilon_{pred}(\mathbf{x}_t^*, t)\|^2 = (1-t)^2 \|v^* - v_{pred}(\mathbf{x}_t^*, t)\|^2. \tag{3}$$

The final Flow-DPO loss [20] is formulated as:

$$-\mathbb{E}\Bigg[ \log \sigma \Bigg( -\frac{\beta_t}{2} \Big( \|v^w - v_\theta(\mathbf{x}_t^w, t)\|^2 - \|v^w - v_{\mathrm{ref}}(\mathbf{x}_t^w, t)\|^2$$
$$- \big( \|v^l - v_\theta(\mathbf{x}_t^l, t)\|^2 - \|v^l - v_{\mathrm{ref}}(\mathbf{x}_t^l, t)\|^2 \big) \Big) \Bigg) \Bigg], \tag{4}$$

where $\beta_t = \beta(1 - t^2)$.

Intuitively, minimizing this loss encourages the model to improve its denoising performance on preferred samples $\mathbf{x}_t^w$ relative to less preferred samples $\mathbf{x}_t^l$ [20, 35]. This guides the predicted velocity field $v_\theta$ to align more closely to videos exhibiting better 3D consistency while diverging from those with poorer geometric coherence.

## 3.2   3D Consistency Metric

We evaluate the 3D consistency of generated videos by validating how well they satisfy epipolar geometry constraints. Epipolar geometry represents the intrinsic projective relationship between two views of the same scene, depending only on the camera's internal parameters and relative positions. In perfectly consistent 3D scenes, corresponding points across different viewpoints must adhere to these geometric constraints.

For any two corresponding points $\mathbf{x}$ in one frame and $\mathbf{x}'$ in another, the epipolar constraint $\mathbf{x}'^T \mathbf{F} \mathbf{x} = 0$ must be satisfied, where $\mathbf{F}$ is the fundamental matrix. This constraint ensures that a point in one view must lie on its corresponding epipolar line in the other view. The fundamental matrix encapsulates the geometric relationship between the two camera poses. It can be formulated as $\mathbf{F} = [\mathbf{e}']_\times \mathbf{P}' \mathbf{P}^+$, where $\mathbf{P}$ and $\mathbf{P}'$ are the camera projection matrices, $\mathbf{P}^+$ is the pseudo-inverse of $\mathbf{P}$, and $\mathbf{e}'$ is the epipole in the second view.

Given a pair of frames $\mathbf{x}_i$ and $\mathbf{x}_j$ from a generated video, we first compute a set of point correspondences using SIFT [42] feature matching. While we validate the method with a simple, robust

handcrafted descriptor, the pipeline can also leverage more recent learned descriptors [43–46]. These correspondences provide a robust set of matching points between the different viewpoints. We then estimate the fundamental matrix using the normalized 8-point algorithm within a RANSAC [47] framework to handle outliers.

Once we have estimated the fundamental matrix, we can measure the geometric consistency using the Sampson epipolar error [21]:

$$S_E = \frac{(\mathbf{x}'^T \mathbf{F} \mathbf{x})^2}{(\mathbf{F}\mathbf{x})_1^2 + (\mathbf{F}\mathbf{x})_2^2 + (\mathbf{F}^T\mathbf{x}')_1^2 + (\mathbf{F}^T\mathbf{x}')_2^2} \tag{5}$$

The Sampson error provides a first-order approximation to the geometric distance between a point and its epipolar line. Lower Sampson error values indicate better adherence to projective geometry constraints and, thus, more consistent 3D structure in the generated videos.

### 3.3 Implementation Details

We conduct experiments with a state-of-the-art open-source video diffusion model called Wan2.1 [4], which possesses 1.3 billion parameters. Our approach is validated in text-to-video and image-to-video generation setups to demonstrate versatility across conditioning types.

**Offline Dataset Generation**    Since our method focuses on 3D-consistent scene generation, we require videos of static scenes with dynamic camera movements. We extract text prompts from the DL3DV [48] and RealEstate10K [49] datasets, provided by [50], containing a wide variety of indoor and outdoor scenes. We generate three videos per caption to ensure sufficient variation in 3D consistency quality, as our preliminary experiments showed that pairs generated from just two samples often lacked meaningful geometric differences. This approach balances computational efficiency with training data quality. We filter put near-static videos to prevent the model from learning a degenerate solution of minimizing camera movement to satisfy epipolar constraint. In total, we generate 24,000 videos for text-to-video and 30,000 videos for image-to-video training, requiring approximately 1,980 GPU hours on NVIDIA A6000s.

**Training Configuration**    Given the computational demands of fine-tuning large video diffusion models, we implement our approach using Low-Rank Adaptation (LoRA) [51] with rank $r = 64$ and $\alpha = 128$. This strategy offers the additional benefit of eliminating the need to store the reference model separately in memory, since the base model with the adapter disabled naturally serves as $p_{\text{ref}}$ during training. We train with a batch size of 32 for 10,000 iterations using the AdamW [52] optimizer with a learning rate of $5 \times 10^{-6}$ and 500 warmup steps. The finetuning takes 2 days on 4 A6000 GPUs.

## 4  Experiments

We assess the effectiveness of our epipolar-aligned video diffusion model compared to baseline approaches and evaluate its impact on scene consistency, visual quality, and prompt alignment. Our evaluation setup consists of 200 videos extracted from the test sets of DL3DV [48] and RealEstate10K [49] datasets, covering a diverse range of indoor and outdoor scenes. To thoroughly test geometric consistency under challenging conditions, we amplify the complexity of camera motion by augmenting prompts with motion-specific phrases (e.g., "orbiting around," "zooming in," "panning across"). We evaluate our model across three complementary benchmarks: (1) the VideoReward benchmark [20], which measures general video generation quality; (2) VBench [53], which provides standardized metrics for temporal consistency and visual fidelity; and (3) our custom suite of 3D consistency metrics based on epipolar geometry constraints. This multi-protocol evaluation approach allows us to comprehensively assess the generated videos' perceptual quality and geometric consistency.

Figure 3 shows some qualitative examples. Before our fine-tuning, the videos often contain morphing objects or inconsistent geometry.

### 4.1  VideoReward Benchmark Evaluation

The VideoReward [20] benchmark evaluates videos across Visual Quality, Motion Quality, and Text Alignment dimensions using a vision language model [27] finetuned on human preferences.

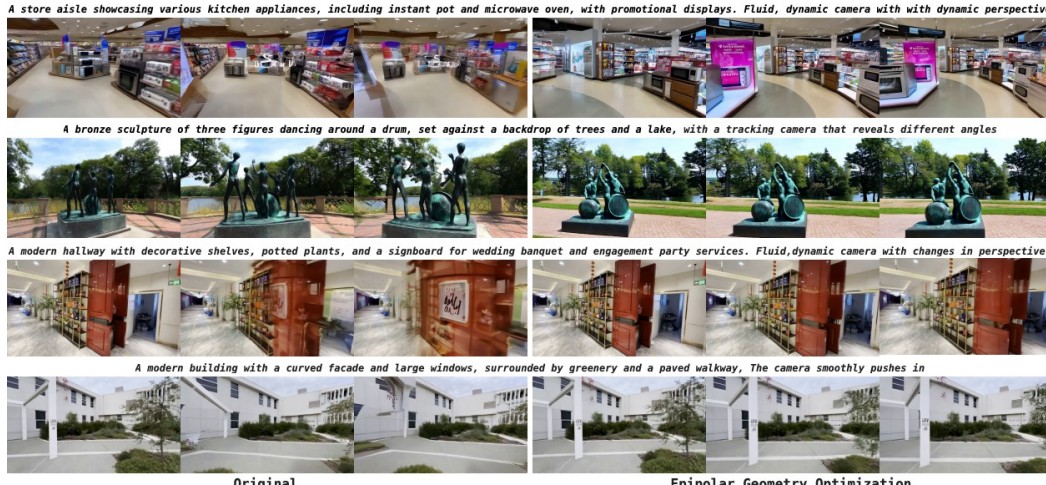

Figure 3: **Qualitative Evaluation:** Visual comparison between the videos generated by the base and finetuned model. First two rows: Wan-2.1-T2V [4], Last two: Wan-2.1-I2V. Our finetuning significantly reduces artifacts and enhances motion smoothness, resulting in more geometrically consistent 3D scenes. Best seen in the supplementary video.

Table 1: **Win-rate vs. original model** on the VideoReward [20] benchmark compared to a learned metric [54].

| Text-to-Video | | | | |
|---|---|---|---|---|
| Method | Visual Quality | Motion Quality | Text Alignment | Overall |
| DPO-MET3R [54] | 56.5% | 64.5% | 44.0% | 55.0% |
| DPO-Epipolar | **72.0%** | **71.0%** | **55.0%** | **73.0%** |
| Image-to-Video | | | | |
| Method | Visual Quality | Motion Quality | Text Alignment | Overall |
| DPO-MET3R [54] | 47.02% | 51.19% | **54.76%** | 48.21% |
| DPO-Epipolar | **51.35%** | **56.08%** | 49.32% | **52.02%** |

Annotators select preferences between video pairs, and a VLM simulates these judgments. We use the resulting pairwise scores to compute win rates of our finetuned model versus the baseline. We also compare against a model aligned with MET3R [54] to assess how our epipolar geometry metric compares to modern 3D vision metrics [55]. Table 1 presents the results of this evaluation. Our text-to-video model significantly outperforms both the baseline and MET3R-based models across all metrics, with win rates of 72.0%, 71.0%, and 55.0% for Visual Quality, Motion Quality, and Text Alignment respectively. This demonstrates that alignment with epipolar constraints enhances not only motion quality but also visual fidelity by reducing artifacts. The image-to-video model, trained with more conservative hyperparameters to minimize baseline deviation, still shows meaningful improvements over both the baseline and MET3R-aligned models in most categories.

## 4.2 VBench Benchmark Evaluation

VBench [53] introduces a comprehensive benchmark suite for video generative models. It consists of a large set of metrics across multiple dimensions, facilitating fine-grained and objective evaluation. We provide the results on five metrics related to visual and motion quality. **Background Consistency** evaluates the temporal consistency of the background scenes by calculating CLIP [56] feature similarity across frames. **Aesthetic Quality** evaluates the artistic and beauty value humans perceive towards each video frame using the LAION aesthetic predictor [30], measuring such concepts as layout and photo-realism. **Temporal Flickering** extracts static frames and computes the mean absolute difference across frames. **Motion Smoothness** validates whether generated motion follows

Table 2: Results on the VBench [53] metrics comparing our epipolar-aligned model against the original model.

| | | Text-to-Video | | | |
|---|---|---|---|---|---|
| Method | Background Consistency | Aesthetic Quality | Temporal Flickering | Motion Smoothness | Dynamic Degree |
| Baseline | 0.930 | 0.541 | 0.958 | 0.981 | **0.815** |
| Ours | **0.942** | **0.551** | **0.969** | **0.984** | 0.627 |
| | | Image-to-Video | | | |
| Method | Background Consistency | Aesthetic Quality | Temporal Flickering | Motion Smoothness | Dynamic Degree |
| Baseline | **0.955** | 0.498 | **0.981** | **0.992** | **0.378** |
| Ours | **0.955** | **0.499** | 0.980 | **0.992** | 0.343 |

Table 3: 3D consistency metrics comparing our epipolar-aligned model against the baseline and MET3R [54] approach.

| | | Text-to-Video | | |
|---|---|---|---|---|
| Method | Motion (mean SSIM) | Perspective Fields | Sampson Distance | MET3R |
| Baseline | 0.233 | 0.426 | 0.190 | 0.050 |
| DPO-MET3R [54] | 0.232 | **0.438** | 0.176 | **0.049** |
| DPO-Epipolar | **0.223** | 0.428 | **0.127** | **0.049** |
| | | Image-to-Video | | |
| Method | Motion (mean SSIM) | Perspective Fields | Sampson Distance | MET3R |
| Baseline | 0.239 | 0.504 | 0.215 | **0.048** |
| DPO-MET3R [54] | **0.220** | **0.517** | 0.202 | 0.049 |
| DPO-Epipolar | 0.239 | 0.515 | **0.197** | 0.049 |

the physical law of the real world. It utilizes the motion priors in the video frame interpolation model [57] to evaluate the smoothness of generated motions. Finally, **Dynamic Degree** employs RAFT [58] to estimate the degree of dynamics in synthesized videos. The results are presented in Table 2. We compare the finetuned Wan-2.1 [4] models to the baseline. The text-to-video model improves the scores across all metrics; however, it sacrifices the dynamic degree by a bit. Nevertheless, the other benchmarks Table 3 and Table 1 demonstrate that the finetuned model generates comparable or superior dynamics. The image-to-video model, being finetuned, reduces the amount of edge cases that perform comparably to the baseline.

## 4.3 3D Geometry Evaluation

Last, we evaluate the direct impact of the finetuned models on 3D geometry metrics. Table 3 shows results across multiple geometric consistency measures. We assess Sampson error (our primary optimization target), MET3R score [54], the realism of Perspective Fields [59] and Motion Level. Perspective Fields classifier [60] evaluates the realism of image perspective fields. Since the metric is image-level, we compute the mean metric across all frames. Additionally, we validate whether the models tend to produce nearly static videos by computing the mean SSIM score between the first and all the other video frames, hence, high scores for static content. Models finetuned with epipolar geometry constraints show significant improvement in Sampson distance (33% reduction in text-to-video), while matching MET3R-optimized models on their own metric. This confirms that classical epipolar geometry provides a cleaner optimization signal than learned metrics, which show only modest self-improvement due to noise when evaluating generated content.

Table 4: Win-rate on the VideoReward [20] benchmark comparing different finetuning strategies.

| Method | Visual Quality | Motion Quality | Text Alignment | Overall |
|---|---|---|---|---|
| sup. finetuning (SFT) | 66.0% | 63.0% | 54.0% | 64.5% |
| Flow-RWR [20] | 63.5% | 60.5% | **57.0%** | 64.0% |
| DRO [36] | 65.0% | 54.0% | 50.5% | 64.5% |
| DPO [17] | **72.0%** | **71.0%** | 55.0% | **73.0%** |

## 4.4 Comparison with Other Fine-tuning Techniques

We compare four finetuning strategies as shown in Table 4: Supervised Finetuning (SFT), Flow-based Reward-Weighted Regression (Flow-RWR) [20], Direct Reward Optimization (DRO) [36], and our proposed DPO with Sampson Error. SFT directly optimizes for minimal epipolar error but struggles without negative samples to distinguish consistency levels. Flow-RWR weights samples by reward values but suffers from inconsistent absolute metrics, while DRO eliminates reference model queries but deviates substantially from the baseline capabilities. Our approach outperforms all alternatives, achieving the highest win rates in Visual Quality (72.0%), Motion Quality (71.0%), and Overall score (73.0%). This demonstrates that preference-based optimization with geometric constraints provides more effective guidance than approaches relying on absolute metrics or unconstrained optimization. Notably, our method achieves these improvements while maintaining the generative flexibility of the original model, allowing it to produce diverse outputs that satisfy both creative and geometric requirements simultaneously.

## 5 Limitations and Broader Impact

Our approach primarily focuses on static scenes with dynamic camera movements, aligning well with applications in 3D reconstruction and novel view synthesis. Adapting this method to scenes with dynamic objects would require modifying the training pipeline to separately model and evaluate object motion and camera movement. Additionally, epipolar geometry constraints assume point correspondences coming from a static scene under camera motion, limiting effectiveness for scenes with independent object movement or non-rigid deformations where a single fundamental matrix cannot explain all correspondences. Video generation models may be misused to produce realistic but deceptive content, contributing to the spread of misinformation, political manipulation, and erosion of public trust. Furthermore, the computational resources required to train such models raise environmental concerns and may exacerbate inequalities in access to advanced AI technologies. Geometry-aware video generation can facilitate various 3D vision tasks, including scene reconstruction, SLAM, and visual odometry. By improving geometric consistency in generated videos, our method produces more realistic and usable synthetic data for training computer vision systems. This advances applications in robotics and autonomous navigation, where accurate spatial understanding is crucial. The integration of classical geometry principles with modern generative models represents a promising direction for enhancing AI systems with stronger physical world understanding.

## 6 Conclusion

We have presented a novel approach for enhancing 3D consistency in video diffusion models by leveraging classical epipolar geometry constraints as preference signals. Our work demonstrates that aligning modern generative models with fundamental geometric principles can significantly improve the spatial coherence of generated content. The robust, mathematically grounded signal from simple Sampson error calculations provides clear guidance without requiring complex 3D supervision or differentiable rewards. The resulting models generate videos with notably fewer geometric inconsistencies and more stable camera trajectories while preserving creative flexibility. This work highlights how classical computer vision algorithms can effectively complement deep learning approaches, addressing limitations in purely data-driven systems and improving generated content quality through adherence to fundamental physical principles.

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
