# OpenReview forum: "Epipolar Geometry Improves Video Generation Models"
_NeurIPS.cc/2025/Conference — Submitted to NeurIPS 2025_

### Official Review · Reviewer_npJ7 · 2025-06-21

**Clarity:** 2
**Significance:** 2
**Originality:** 3
**Rating:** 4
**Confidence:** 4

**Summary:**

This paper proposes to improve the 3D consistency of video generation through the epipolar constraint, while the optimization is based on reinforcement learning, i.e., diffusion (flow-matching) DPO. Specifically, they employ the SIFT feature matching to achieve the fundamental matrix, and then use the Sampson epipolar error as the reward. The fine-tuning is implemented with LoRA. Comprehensive experiments and qualitative videos submitted in the supplementary show the effectiveness of the proposed method.

**Questions:**

My current inclination is to vote for the acceptance of this paper, though I still have significant concerns. These concerns are primarily rooted in the first three weaknesses, which critically informed my final assessment.

**Ethical Concerns:**

["NO or VERY MINOR ethics concerns only"]

**Final Justification:**

I retain my score of borderline accept. This paper presents innovative ideas for generating 3D consistent videos. But it would be very helpful to the community if the authors could further provide the results of controlled video generation (like camera control). To my knowledge, some camera control methods like Uni3C, frozen their backbone modules and only a control branch is trained, which can be seamlessly integrated into the geometry-aligned model. Note that the concern about dynamic degree and controllability still remain because these results are not mentioned in this paper, while the dynamic degree is actually reduced as shown in the paper's experiments.

**Limitations:**

Yes.

**Paper Formatting Concerns:**

No formatting concerns are included in this paper.

**Quality:**

3

**Strengths And Weaknesses:**

Strengths:
1. The idea of leveraging 3D consistent metrics to align the video model through DPO is interesting.
2. This paper includes comprehensive experiments, such as the VideoReward benchmark, VBench, 3D consistency metrics, and the comparison among different fine-tuning strategies.
3. The paper is well-organized, while the Sampson epipolar error used in this work is convincing.

Weaknesses:
1. Since this paper uses the DPO learning, which addresses the non-differentiable issue, more heuristic metrics should also be discussed and compared. For example, as this paper utilized SIFT and feature matching to get the fundamental matrix, many other matching metrics should be considered (number of matching points, SfM points,  matching confidence, and re-projection error), while the epipolar error is just a special case of them. This paper lacks these discussions to claim the superiority of the epipolar error.
Since DPO does not care about the reward value, the authors should provide more evidence to demonstrate that the Sampson epipolar error is the best choice.

2. Another concern is whether the proposed epipolar constraint will potentially hinder the dynamicity of the video model, as shown in Table 2. Intuitively, smaller movements lead to better 3D consistency. However, the authors did not include any penalty for small camera movement, which may make the model tend to synthesize videos with small but consistent camera motions.

3. Why training and evaluation are limited only on DL3DV and Real10K? Theoretically, one advantage of reinforcement learning is that it can learn any results generated by itself without the limitation of real-world datasets, resulting in strong generalization across various domains. However, this work only focuses on real-world multi-view images, which is very confusing. Furthermore, the authors should provide VBench results based on natural videos to show that the generalization of the LoRA fine-tuned model is preserved.

4. Some introductions about the Diffusion/Flow-DPO are not sufficient. The meaning of $\sigma$ in Eq(2), Eq(4) is not mentioned. When the DPO is starting ($v_{ref}=v_{\theta}$), the loss definition would suffer from a numerical issue.

5. Some presentations of the figures should be improved. The epipolar lines drawn in Figures 1 and 2 are confusing. I suggest highlighting the epipolar "error" between the source points and target lines in the figures across different generations.

---

> ### Author Rebuttal · Authors · 2025-07-30
>
> We thank the reviewer for recognizing the effectiveness of our approach and the comprehensive nature of our experiments. We appreciate the positive feedback on leveraging 3D consistency metrics for DPO alignment.
>
> **Q1:**
> > Since this paper uses the DPO learning, which addresses the non-differentiable issue, more heuristic metrics should also be discussed and compared. For example, as this paper utilized SIFT and feature matching to get the fundamental matrix, many other matching metrics should be considered (number of matching points, SfM points, matching confidence, and re-projection error), while the epipolar error is just a special case of them. This paper lacks these discussions to claim the superiority of the epipolar error. Since DPO does not care about the reward value, the authors should provide more evidence to demonstrate that the Sampson epipolar error is the best choice.
>
> We do not claim that the Sampson epipolar error is the best choice of all classical metrics. For example, there are nearly infinite permutations of keypoint detectors, descriptors, and matchers, out of which some might perform better under specific conditions. Instead, we make a more general claim: classical, mathematically grounded geometric constraints provide more reliable preference signals than learned or sophisticated alternatives for video model alignment. Our experiments consistently show that even simple SIFT + Sampson error outperforms advanced descriptors and learned metrics, demonstrating that mathematical simplicity and stability are more valuable than sophistication for effective DPO optimization.
> Regarding alternative matching metrics, we refer to **rDPn Q5**, where we compare SIFT, LightGlue, and SEA-Raft descriptors, and **gMLo Q1**, where we evaluate symmetric epipolar error.
>
> **Q2:**
> > Another concern is whether the proposed epipolar constraint will potentially hinder the dynamicity of the video model, as shown in Table 2. Intuitively, smaller movements lead to better 3D consistency. However, the authors did not include any penalty for small camera movement, which may make the model tend to synthesize videos with small but consistent camera motions.
>
> Yes, this is a valid concern. While we partially mitigate it through filtering near-static videos (lines 215-217), simpler motions naturally have lower epipolar error. For the rebuttal, we experimented with explicit static content penalty and find that a simple temporal variation penalty with small $\lambda$ prevents degradation to near static scenes.
>
> We add a temporal penalty term:
> $$L_{temporal} = -\lambda \cdot E[Var(x_{clean})]$$
>
> where $x_{clean} = x_{noisy} - \sigma \cdot f_\theta(x_{noisy}, t)$ and the variance is computed across the temporal dimension.
>
> | Method | Dynamic Degree | Motion Quality | Sampson Error |
> |--------|----------------|----------------|---------------|
> | Ours | 0.627 | 71.5% | 0.127 |
> | Ours + Static Penalty | 0.710 | 69.5% | 0.131 |
>
> This approach maintains geometric consistency while preserving motion dynamics, addressing the trade-off between 3D consistency and video dynamicity.
>
> **Q3:**
> > Why training and evaluation are limited only on DL3DV and Real10K? Theoretically, one advantage of reinforcement learning is that it can learn any results generated by itself without the limitation of real-world datasets, resulting in strong generalization across various domains. However, this work only focuses on real-world multi-view images, which is very confusing. Furthermore, the authors should provide VBench results based on natural videos to show that the generalization of the LoRA fine-tuned model is preserved.
>
> We would like to clarify that we only use the captions for these datasets and not the videos themselves.
> (Please also see our response to **hTo3 Q3**) - we use these for reliable geometric constraints in static scenes with dynamic cameras. Thank you for the generalization suggestion. We extended evaluation to both VBench 2.0 and VideoReward with challenging general prompts:
>
> **VideoReward Metrics:**
> | Benchmark | Visual Quality | Motion Quality | Text Alignment | Overall |
> |-----------|----------------|----------------|----------------|---------|
> | VBench 2.0 | 61.3% | 55.3% | 52.0% | 57.9% |
> | VideoReward | 65.0% | 58.5% | 50.5% | 58.5% |
>
> This shows that using prompts for static videos with moving cameras is enough to learn a generalizable model.
>
> **VBench Metrics:**
> | Model | Background Consistency | Aesthetic Quality | Temporal Flickering | Motion Smoothness | Dynamic Degree |
> |-------|------------------------|-------------------|---------------------|-------------------|----------------|
> | Baseline | 0.9517 | 0.535 | 0.979 | 0.986 | 0.595 |
> | Ours | 0.9538 | 0.541 | 0.983 | 0.989 | 0.557 |
>
> Our key insight from these evaluations is that learning general geometrical principles also often improves physical stability and object interactions (e.g., person walking with surfboard: baseline shows flickering surfboard piercing through body, while ours maintains physical stability). We will add these experiments and examples to the final version.
>
> **Q4:**
> > Some introductions about the Diffusion/Flow-DPO are not sufficient. The meaning of $\sigma$ in Eq(2), Eq(4) is not mentioned. When the DPO is starting ($v{ref}=v{\theta}$), the loss definition would suffer from a numerical issue.
>
> We will improve the objective function description and clarify the notation. The confusion arises because we accidentally omitted stating that $\sigma$ is the sigmoid function. When DPO starts with $v_{\text{ref}} = v_\theta$, the loss becomes $-\log(\sigma(0)) = -\log(0.5)  \approx 0.693$, which provides a well-defined starting point without numerical issues. We will add this clarification and expand the Flow-DPO background in the revision.
>
> **Q5:**
> > Some presentations of the figures should be improved. The epipolar lines drawn in Figures 1 and 2 are confusing. I suggest highlighting the epipolar "error" between the source points and target lines in the figures across different generations.
>
> Yes, thank you for this suggestion! We will improve the figure presentations by highlighting the epipolar error between source points and target lines across different generations to better demonstrate the geometric inconsistencies and how our method reduces them.

---

> > ### Comment · Reviewer_npJ7 · 2025-08-02
> > **Questions about the dynamic generalization**
> >
> > Thanks for the rebuttal from the authors. Some of my concerns are addressed. While some of my concerns have been addressed, generalization remains the primary issue. Preserving 3D consistency is crucial for video generation and can be examined under two distinct scenarios: 1) Consistency in general video generation: This includes applications like Veo3, Keling2.1, Seedance, etc. 2) Consistency in controlled video generation: Specifically, scenarios requiring camera control.
> >
> > General video generation: For this scenario, the proposed method must ensure that the foundational video models (e.g., Wan, CogvideoX) retain their ability to generate dynamic motion and inherent variability. The fine-tuned model should demonstrate it can maintain multi-view consistency for static objects while allowing for natural motions, such as human movements.
> >
> > Controlled video generation: In cases where controllability (e.g., camera control) is essential, the proposed method is especially significant. However, its success depends upon maintaining this controllability. Sacrificing controllability to achieve consistency would be counterproductive and reduce its utility.
> >
> > I hope to see more details or evidence that the proposed method can handle these scenarios. Otherwise, the practical impact of this approach remains limited.

---

> > > ### Author Response · Authors · 2025-08-04
> > > **Addressing additional questions about generalization**
> > >
> > > Thank you for acknowledging that we have addressed most concerns and for highlighting the importance of 3D consistency in video generation. We agree that generalization is crucial for adoption. Our choice of DPO alignment specifically supports this: DPO generalizes better than supervised fine-tuning through implicitly enforcing KL-regularization, preserving base model capabilities [1, 2].
> > > Below, we have summarized all our experiments that demonstrate generalization and controllability.
> > >
> > > > General video generation
> > >
> > > Our experiments on the open-source Wan-2.1-1.3B model demonstrate preserved generalization:
> > > * Supplementary Table 1 shows our aligned model generalizes to dynamic scenes, including human movements from MiraData-9K. The evaluation on general prompts of dynamic scenes shows 58.5\% overall quality win rate, demonstrating effectiveness beyond static training scenes.
> > > * Our experiments on VideoReward and VBench benchmarks (rebuttal Q3) demonstrate improvements across all metrics on diverse prompts, with only minor Dynamic Degree reduction on VBench due to edge cases in near-static scenes (like near-static scene of trees swaying in wind becoming static, while maintaining substantial improvements in other metrics).
> > > * We observe a positive impact on physical stability in dynamic scenes. As we are now allowed to show samples here, we will include them in the final revision.
> > >
> > > > Controlled video generation
> > >
> > > ### Text-based Camera Control
> > > * Table 1 demonstrates preserved controllability on static scenes with camera motion captions. Our method achieves 55.0\% overall win rate on camera motion prompts ("orbiting around," "panning across"), demonstrating preserved and enhanced controllability for static scenes with dynamic cameras.
> > > * Supplementary results (Table 1) and rebuttal experiments (Q3) also show superior text alignment across diverse static scenes.
> > > ### Explicit Camera Control Mechanisms
> > > * This is an active research area and orthogonal to our work: it can benefit from geometry-aligned models. We expect modules like Uni3C [3] could be trained on our pre-aligned model in the following order: Base Model → Geometry Alignment → Add Camera Control. Since we only improve base capabilities without changing architecture, this facilitates precise control module development. We will release aligned checkpoints to support this.
> > >
> > > [1] Rafailov, Rafael, et al. "Direct preference optimization: Your language model is secretly a reward model." Advances in neural information processing systems 36 (2023): 53728-53741.
> > >
> > > [2] Wallace, Bram, et al. "Diffusion model alignment using direct preference optimization." Proceedings of the IEEE/CVF Conference on Computer Vision and Pattern Recognition. 2024.
> > >
> > > [3] Cao, Chenjie, et al. "Uni3C: Unifying Precisely 3D-Enhanced Camera and Human Motion Controls for Video Generation." arXiv preprint arXiv:2504.14899 (2025).

---

> > > > ### Comment · Reviewer_npJ7 · 2025-08-06
> > > > **Response**
> > > >
> > > > Thanks for the reply. I retain my score. This paper presents innovative ideas for generating 3D consistent videos. But it would be very helpful to the community if the authors could further provide the results of controlled video generation (like camera control). To my knowledge, some camera control methods like Uni3C, frozen their backbone modules and only a control branch is trained, which can be seamlessly integrated into the geometry-aligned model. Note that the concern about dynamic degree and controllability still remain because these results are not mentioned in this paper, while the dynamic degree is actually reduced as shown in the paper's experiments.

---

> > > > > ### Author Response · Authors · 2025-08-06
> > > > > **Response**
> > > > >
> > > > > Thank you for the productive discussion!
> > > > >
> > > > > Regarding controlled video generation, we agree that this represents an important direction. Camera control methods are typically trained on top of a frozen base model, which could also be easily done for our aligned checkpoint. However, this falls outside our paper's scope, as it is a downstream task.
> > > > >
> > > > > We agree that the dynamics-consistency tradeoff is crucial, which is why we evaluate across multiple metrics: VBench Dynamic Degree (RAFT-based), VideoReward Motion Quality (human-distilled VLM), mean SSIM similarity, and human preference scores. Single neural network metrics can be biased, so our multi-metric evaluation demonstrates the tradeoff while showing that we preserve the overall motion quality.

---

### Official Review · Reviewer_gMLo · 2025-07-03

**Clarity:** 2
**Significance:** 3
**Originality:** 3
**Rating:** 4
**Confidence:** 5

**Summary:**

This work explores using epipolar geometry constraints to post‐train latent video diffusion models for enhanced 3D consistency. The main contributions are:

-Leveraging pairwise Sampson error as a reward in a Direct Preference Optimization (DPO) to rank and fine‐tune generated video samples for improved geometric coherence
-Introducing a classical epipolar errornthat requires only relative comparisons, thereby avoiding the need for explicit 3D supervision
-Constructing a dataset of over 162 K generated video pairs annotated with epipolar consistency metrics for downstream alignment and evaluation.

**Questions:**

see comments above

**Ethical Concerns:**

["NO or VERY MINOR ethics concerns only"]

**Final Justification:**

Thank you for the detailed explanation in your response. The authors successfully addressed most of the questions and provided additional details, comparisons, and results. I agree it's an interesting work to explore epipolar geometry in video generation, and would like to raise the original score.

**Limitations:**

see comments above

**Paper Formatting Concerns:**

format is good

**Quality:**

2

**Strengths And Weaknesses:**

Strengths
- Employing epipolar constraints as a preference signal bridges deep generative models with mathematically grounded 3D principles
- Evaluation spans perceptual (VideoReward, VBench) and geometry‐specific (Sampson error, Perspective Fields) measures

Major Weaknesses
-While Sampson error is effective, the paper lacks comparison to other geometric criteria (e.g., symmetric epipolar error, learned multi‐view consistency scores) to justify its optimality.
-Details on the human‐in‐the‐loop VideoReward comparisons (e.g., number of raters, agreement levels) are sparse.
-The preference dataset and evaluation rely exclusively on synthetic videos generated by the same base model (Wan-2.1). Without validation on external datasets or unseen models, it’s unclear if the approach generalizes beyond this
-Key methodological details (e.g., choice of DPO hyperparameters, criteria for video filtering) are scattered or missing.
-The authors should report results for the base video diffusion model (e.g., Wan-2.1) without any epipolar fine-tuning. Also a control variant that injects the same number of LoRA parameters but uses a zero-reward DPO. This will demonstrate that performance gains stem specifically from the geometry signal rather than simply adding parameters or preference tuning.
-Provide timing benchmarks breaking down diffusion sampling, epipolar error computation, and DPO update on a single GPU. Compare this end-to-end runtime to a one-pass LoRA baseline, so readers can assess the practical trade-off between improved consistency and added cost.

---

> ### Author Rebuttal · Authors · 2025-07-30
>
> We thank the reviewer for recognizing the value of bridging deep generative models with mathematically grounded 3D principles. We appreciate the acknowledgment of our comprehensive evaluation spanning both perceptual and geometry-specific measures. Below we address each concern in detail:
>
> **Q1:**
> > While Sampson error is effective, the paper lacks comparison to other geometric criteria (e.g., symmetric epipolar error, learned multi‐view consistency scores) to justify its optimality.
>
> We appreciate this suggestion and have conducted evaluations on alternative geometric criteria. We refer to **rPDn Q5** where we compare SIFT, LightGlue, and SEA-Raft descriptors, and to our Table 3 MET3R experiments demonstrating that learned multi-view consistency scores are too noisy for effective alignment (training with MET3R scores does not improve MET3R). We also evaluate symmetric epipolar error, visually observing that the difference between models trained on these two metrics is rather insignificant:
>
> | Metric | Visual Quality | Motion Quality | Text Alignment | Overall |
> |--------|----------------|----------------|----------------|---------|
> | Sampson Error | 64.3% | 64.2% | 41.8% | 57.1% |
> | Symmetric Epipolar Error | 76.4% | 59.6% | 36.4% | 56.4% |
>
> To further validate our approach, we conducted a cross-metric evaluation showing that models trained with geometric constraints generalize better than VLM-based training, which provides too noisy rankings:
>
> | Train Metric | Test: VideoReward Motion Quality | Test: Sampson Error |
> |--------------|----------------------------------|---------------------|
> | Sampson | 64.3% | 0.127 |
> | VideoReward Motion Quality | 61.3% | 0.179 |
>
> While there are many possible metric configurations, our core message is that simplicity and reliability outperform learned or state-of-the-art metrics: even a basic SIFT + Sampson setup surpasses VLM-based or foundational model metrics. We would appreciate more specific references if the reviewer has other particular geometric criteria in mind that we should add to the final version. Overall, our experiments show that classical, mathematically grounded approaches provide more stable optimization signals than complex learned alternatives.
>
> **Q2:**
> > Details on the human‐in‐the‐loop VideoReward comparisons (e.g., number of raters, agreement levels) are sparse.
>
> We use the pre-trained VideoReward vision-language model for automatic evaluation, not direct human annotation. VideoReward authors previously distilled human preference scores into a VLM that simulates human judgments across Visual Quality, Motion Quality, and Text Alignment dimensions. We apply this trained model to automatically rate our generated videos and compute win rates against the baseline, following the standard VideoReward benchmark protocol. This approach provides consistent, scalable evaluation without requiring additional human raters.
>
> **Q3:**
> > The preference dataset and evaluation rely exclusively on synthetic videos generated by the same base model (Wan-2.1). Without validation on external datasets or unseen models, it’s unclear if the approach generalizes beyond this
>
> We are learning a LoRA module on top of an existing video generator. This cannot generalize to unseen models as it is specific to the base model.
> DPO alignment inherently requires generating preference data from the same base model being fine-tuned. However, we do evaluate generalization beyond the training domain through out-of-domain scenarios and multiple evaluation frameworks. While computational constraints (1,980 GPU hours for training data generation) prevent multi-model validation, our approach is fundamentally model-agnostic, relying on standard DPO training applicable to any video diffusion model. Please also see our response to **rPDn Q4**.
>
>
> **Q4:**
> > Key methodological details (e.g., choice of DPO hyperparameters, criteria for video filtering) are scattered or missing
>
> We refer to **hTo3 Q1** where we expand on our video filtering criteria: we only use training pairs where $(\text{metric}(x_{\text{win}}) - \text{metric}(x_{\text{lose}}) > \tau) \wedge (\text{metric}(x_{\text{win}}) > \epsilon)$, ensuring genuine consistency gaps. For DPO hyperparameters, we use established best practices with conservative settings (learning rate $5 \times 10^{-6}$, LoRA rank 64, $\alpha=128$) to prevent overfitting. We will add a comprehensive paragraph on hyperparameters and key methodological details to the supplementary materials, including filtering thresholds, training configurations, and dataset statistics. Our code and models will be released for full reproducibility.
>
> **Q5:**
> > The authors should report results for the base video diffusion model (e.g., Wan-2.1) without any epipolar fine-tuning. Also, a control variant that injects the same number of LoRA parameters but uses a zero-reward DPO. This will demonstrate that performance gains stem specifically from the geometry signal rather than simply adding parameters or preference tuning
>
> We do indeed provide results for the base video diffusion model ("Baseline" rows in all tables), and Table 1 shows win rates versus the original model without any fine-tuning. Table 4 compares DPO against other fine-tuning strategies, proving the efficiency of our geometric signal. We will make this clearer in the paper. Importantly, our model learns from negative samples - the gap between consistent and inconsistent trajectories enables learning general geometric principles.
>
> **Q6:**
>
> > Provide timing benchmarks breaking down diffusion sampling, epipolar error computation, and DPO update on a single GPU. Compare this end-to-end runtime to a one-pass LoRA baseline, so readers can assess the practical trade-off between improved consistency and added cost.
>
> Our approach involves two offline stages: (1) preference data generation requiring diffusion sampling and epipolar error computation, and (2) DPO fine-tuning. All ranking happens offline during training data preparation. At inference, our method adds no computational overhead compared to any LoRA-based fine-tuning approach.
>
> | Component | Time (Single Iteration) | Overhead |
> |-----------|------|----------|
> | Wan-2.1 DiT (1.42B) | 25.13s | - |
> | With LoRA (47.19M params) | 25.37s | +0.95% |
> | Epipolar computation (per video) | 0.172s | - |
>
> The meaningful comparison is total training cost: our approach requires ~1,980 GPU hours for dataset generation plus 2 days on 4 A6000s for DPO fine-tuning, versus standard LoRA fine-tuning, which only needs the latter. The geometric consistency improvements come at the cost of additional offline data generation, not runtime inference overhead.

---

> > ### Comment · Area_Chair_Sz56 · 2025-08-06
> > **Author-reviewer Discussion**
> >
> > Dear reviewer,
> >
> > The system shows that you have not yet posted a discussion with the authors. As per the review guidelines, we kindly ask you to evaluate the authors’ rebuttal to determine whether your concerns have been sufficiently addressed before submitting the Mandatory Acknowledgement. If they have, please confirm with the authors. Otherwise, feel free to share any remaining questions or concerns.
> >
> > Thank you for your time and valuable feedback.
> >
> > Your AC

---

> > ### Author Response · Authors · 2025-08-07
> > **Following Up on Our Rebuttal Responses**
> >
> > Thank you for thoughtful feedback on our work. In our rebuttal, we have addressed several areas of uncertainty and provided additional experimental evidence to respond to your questions. As the discussion phase approaches its end, we look forward to your feedback. Having reviewed our rebuttal, are there any outstanding concerns or questions you'd like to discuss? Please feel free to bring up any points or request additional clarification.

---

### Official Review · Reviewer_rPDn · 2025-07-03

**Clarity:** 3
**Significance:** 3
**Originality:** 3
**Rating:** 4
**Confidence:** 4

**Summary:**

This paper presents an attempt to use traditional geometrical constraints to perform post-training fine-tuning of video diffusion models. Specifically, it studies using epipolar line constraints to enforce geometrical consistency in the generated videos by using DPO. Experiments are conducted to show the effectiveness of the proposed framework.

**Questions:**

- The baseline results from Wan, as shown in the supplementary materials, look too bad to me. My impression is that Wan can generate fairly good videos. Are the authors cherry-picking failure cases from Wan to showcase the advantages? Please elaborate on the comprehensive policy on selecting demos.
- L230: How are extracted videos used to evaluate generated content? More details should be elaborated here.

- Some minor comments:
  - All image resolutions in the figures are pretty low. Maybe there are some issues with PDF compression.
  - L167: broken sentences?

**Ethical Concerns:**

["NO or VERY MINOR ethics concerns only"]

**Final Justification:**

The rebuttal has addressed most of my initial concerns. I will keep a positive score.

**Limitations:**

The epipolar constraints assume the camera intrinsics are not distorted. This prevents the video generative models from generating videos like egocentric ones.

**Quality:**

3

**Strengths And Weaknesses:**

**Strengths**

- This paper pivots on a very important direction of enforcing physical inductive biases in video generative models. The presented approach is simple, effective, and clear, and shows a strong message of the importance of physical priors in finetuning video models. Many potential follow-ups can be inspired by this work, including adding even more physical non-differentiable biases to the video models, including physical stability, energy conservation, etc.
- The contributed dataset might be useful in other areas, like aligning or evaluating VLLMs with physical plausibilities.

**Weaknesses**:

- The improvement over quantitative 3D consistency metrics is not significant, especially for image-to-video models. For MET3R metrics, the proposed models are even worse than the baseline model.
- 3D consistency metrics should also be evaluated for Table 4 to provide a more comprehensive study on different fine-tuning strategies.
- Only comparing metrics like Sampson distance or MET3R is not enough to evaluate the 3D consistency. 3D reconstruction should be performed on the generated videos (like, reconstructing a NeRF or 3DGS), and metrics including reprojection error should be evaluated. Both qualitative and quantitative results should be reported on the 3D reconstruction quality.
- Only one model (Wan 2.1) is evaluated in the experiments. More models should be evaluated to show the generalizability of the proposed framework.
- It is not clear how the selection of feature extractors would influence the results. If using feature extractors like SuperGlue or Loftr, would the results be better?
- It is not clear how the proposed framework compares against methods like VPO (Cheng et al., VPO: Aligning Text-to-Video Generation Models with Prompt Optimization).

---

> ### Author Rebuttal · Authors · 2025-07-30
>
> We thank the reviewer for recognizing the importance of incorporating physical inductive biases into video generative models and for highlighting our approach's effectiveness and clarity. We appreciate the acknowledgment that our work demonstrates the value of physical priors and could inspire future research incorporating additional non-differentiable constraints. We also thank the reviewer for suggesting valuable improvements to our evaluation. Below we address each concern:
>
> **Q1.1:**
> > The improvement over quantitative 3D consistency metrics is not significant, especially for image-to-video models.
>
> We observe that image-to-video models show smaller improvements because the conditioning frame inherently constrains them, limiting the geometric inconsistencies that can be corrected. If the conditioning image is geometrically sound, the model inherently performs better. The metric gap between video pairs is smaller for I2V than T2V because videos generated from images are already more consistent than those generated from text.
>
> **Q1.2:**
> > For MET3R metrics, the proposed models are even worse than the baseline model.
>
> Regarding MET3R, this validates one of our core claims: models aligned with MET3R actually fail to improve on their own target metric (Table 3), confirming that learned metrics like MET3R (based on DUST3R) produce noisy signals that compromise alignment quality. This is precisely why we advocate for classical geometric constraints - simple, reliable signals from classical metrics outperform modern learning-based methods that provide unstable optimization targets. We will clarify this and rework Table 3 to highlight better how MET3R's self-improvement failure supports our approach of using mathematically grounded Sampson error for stable, effective alignment.
>
> **Q2:**
> > 3D consistency metrics should also be evaluated for Table 4 to provide a more comprehensive study on different fine-tuning strategies.
>
> Thanks for the suggestion. We did compute the 3D consistency metrics and will extend the Table 4 in the final version.
>
> | Method | Perspective ↑ | Sampson ↓ | Dynamics ↓ |
> |--------|---------------|-----------|------------|
> | Baseline | 0.426 | 0.190 | 0.233 |
> | SFT | 0.427 | 0.161 | 0.225 |
> | RWR | 0.434 | 0.174 | 0.229 |
> | DRO | 0.410 | 0.068 | 0.195 |
> | Ours | 0.428 | 0.127 | 0.223 |
>
> This also demonstrates an importance of evaluation on different set of metrics, for example while DRO achieves even lower Sampson Error, since it doesn't include KL-Divergence term the model produce clear significant visual artifacts which is not captured by only consistency metrics.
>
> **Q3:**
> > Only comparing metrics like Sampson distance or MET3R is not enough to evaluate the 3D consistency. 3D reconstruction should be performed on the generated videos (like, reconstructing a NeRF or 3DGS), and metrics including reprojection error should be evaluated. Both qualitative and quantitative results should be reported on the 3D reconstruction quality.
>
> Yes, this is complementary to our evaluation, and we appreciate the suggestion. We will conduct 3D reconstruction evaluation for the final version using the following pipeline: COLMAP for structure-from-motion, followed by Splatfacto training (3DGS). While we could not complete this evaluation within the rebuttal timeframe due to the time-consuming setup of the entire pipeline but have seen some preliminary positive results.
>
> **Q4:**
> > Only one model (Wan 2.1) is evaluated in the experiments. More models should be evaluated to show the generalizability of the proposed framework
>
> We acknowledge this limitation due to significant computational constraints - generating the 54,000 training videos required approximately 1,980 GPU hours, preventing multi-model evaluation within the rebuttal timeframe. We estimate that repeating our experiments for another video diffusion model would need several weeks of computing time. However, our approach is fundamentally model-agnostic, and all modern latent diffusion models share similar high-level architectures. Thus, we expect that the method will generalize well.
>
> **Q5:**
> > It is not clear how the selection of feature extractors would influence the results. If using feature extractors like SuperGlue or Loftr, would the results be better?}
>
> Thanks. We conducted this evaluation using 600 videos (scaled from 200) with more complex motion captions. We tested LightGlue as a learned descriptor and SEA-Raft for dense correspondences. For computational efficiency, we compute VideoReward VLM metrics:
>
> | Descriptor | Visual Quality | Motion Quality | Text Alignment | Overall |
> |------------|----------------|----------------|----------------|---------|
> | SIFT | 64.3% | 64.2% | 41.8% | 57.1% |
> | LightGlue | 70.3% | 52.6% | 38.5% | 53.8% |
> | SEA-Raft | 80.3% | 56.0% | 33.6% | 56.9% |
>
> Counter-intuitively, better descriptors do not guarantee better alignment. Learned descriptors tend to favor oversaturated scenes, leading to reward hacking for visual quality. More critically, when videos contain artifacts in some regions, advanced descriptors like LightGlue still find good correspondences in clean areas, filtering out correspondences due to bad geometry, resulting in misleadingly low epipolar error. We want correspondences across the entire scene, so artifacts anywhere produce high error. Simpler descriptors provide more robust rankings with wider preference gaps - essentially, state-of-the-art descriptors are "too good" for ranking, missing global geometric inconsistencies that simpler methods reliably detect.
>
> **Q6:**
> > It is not clear how the proposed framework compares against methods like VPO (Cheng et al., VPO: Aligning Text-to-Video Generation Models with Prompt Optimization).
>
> Yes, thank you for the suggestion. We compare against VPO (Video Prompt Optimization), which is complementary to our approach since it optimizes prompts rather than model weights. We evaluate VPO alone and VPO combined with our method:
>
> | Method | Visual Quality ↑ | Motion Quality  ↑| Overall ↑ | Dynamic Degree ↑ | Motion (mean SSIM)↓ |
> |------------|----------------|----------------|---------|----------------|--------------------|
> | Ours | 63.1% | 65.8% | 59.1% | 0.8 | 0.211 |
> | VPO | 59.1% | 70.6% | 82.7% | 0.65 | 0.235 |
> | VPO + Ours | 67.0% | 71.9% | 83.6% | 0.61 | 0.234 |
>
> We observe that VPO tends to reduce camera motion and restructure the prompt while optimizing for general video quality. However, such prompt optimization methods can still be efficiently combined with geometry-aligned models to achieve both high visual quality and geometric consistency.
>
> **Q7:**
> > The baseline results from Wan, as shown in the supplementary materials, look too bad to me. My impression is that Wan can generate fairly good videos. Are the authors cherry-picking failure cases from Wan to showcase the advantages? Please elaborate on the comprehensive policy on selecting demos.
>
> We refer the reviewer to our response in **hTo3 Q4**, where we address this concern with human evaluation results. To summarize: we used the Wan-2.1 1.3B checkpoint due to computational constraints, and while Wan can produce high-quality videos, we observe it still often struggles with significant camera motion, which motivates our approach. Our human evaluation demonstrates that the baseline produces consistent videos only 54.1% of the time. Our method preserves quality for already-consistent videos while dramatically improving inconsistent ones (60.4% vs 7.5% win rate), confirming we target geometric issues. We will expand the demo and supplementary materials with more diverse samples to provide a more comprehensive view of both baseline and improved performance.
>
> **Q8:**
> > How are extracted videos used to evaluate generated content? More details should be elaborated here.
>
> We apologize for the confusion and typo - we will fix this in the revision. We use captions from the RealCam-Vid dataset (which provides captions for DL3DV and RealEstate10K scenes) rather than the videos themselves. These text prompts are used to generate videos with both baseline and fine-tuned models for evaluation. The 200 evaluation videos are generated from diverse prompts covering indoor and outdoor scenes, with motion-specific phrases added (e.g., "orbiting around," "panning across") to create challenging camera movements for testing geometric consistency. We will clarify this in the paper.
>
> **Q9:**
> > The epipolar constraints assume the camera intrinsics are not distorted. This prevents the video generative models from generating videos like egocentric ones.
>
> This is a valid limitation that we will add to our limitations section. Epipolar geometry assumes pinhole cameras without significant distortion, limiting applicability to fisheye or extreme wide-angle egocentric videos. However, most video generation models are trained on internet-scale data with predominantly standard perspective cameras. Our method targets this common case and successfully improves geometric consistency for the vast majority of content. Also, this issue could be partially mitigated through rectification/unwarping, and the framework could potentially be extended using more general geometric constraints for specialized applications.

---

> > ### Comment · Reviewer_rPDn · 2025-08-05
> >
> > Thanks for your rebuttal. I'm keeping my positive score.

---

> ### Author Response · Authors · 2025-08-06
> **Response**
>
> Thank you for the response to our rebuttal and valuable suggestions on model evaluation.
>
> We have now completed the 3D reconstruction evaluation using Splatfacto training (3DGS), with results showing clear improvements:
>
> | Method | PSNR ↑ | SSIM ↑ | LPIPS ↓ |
> |--------|--------|--------|---------|
> | Baseline (Wan-2.1) | 22.32 | 0.7063 | 0.3434 |
> | Ours | 23.13 | 0.7290 | 0.3154 |
>
> The results confirm that improved geometric consistency translates to better downstream 3D reconstruction quality.

---

### Official Review · Reviewer_hTo3 · 2025-07-03

**Clarity:** 3
**Significance:** 2
**Originality:** 3
**Rating:** 4
**Confidence:** 4

**Summary:**

This paper uses classical epipolar geometry constraints as the rule to guide the video generation model.
It first uses Sampson error calcualations to rank the generated videos.
Then, DPO is used to further optimize the video generation models.
After the procedures, the models can produce videos with better geometric consistency.

**Questions:**

See weaknesses

**Ethical Concerns:**

["NO or VERY MINOR ethics concerns only"]

**Final Justification:**

Most of my concerns have been addressed.

**Limitations:**

yes

**Quality:**

2

**Strengths And Weaknesses:**

Strengths
1. The use of epipolar geometry constraints as a guiding principle for DPO is intriguing. This approach is both intuitive and promising.
2. The experiments demonstrate enhanced geometric consistency.

Weaknesses
1. The accuracy of the Sampson epipolar error might not be sufficient. Is it adequate for DPO? What if the two generated videos both have low epipolar accuracy?
2. Could the selection process introduce bias? For instance, some scenes might exhibit high epipolar accuracy while others show lower accuracy. This could lead to category bias after training.
3. I'm curious why the training was conducted on the DL3DV and RealEstate10K datasets instead of general videos. Wouldn't this limit the model's generalization ability?
4. What if the original methods fail to produce geometrically aligned videos? Can this method still offer improvements?
5. From the attached results, I'm curious why the original video quality appears so low. Based on the reviewer's experience, Wan-2.1-T2v typically delivers much better video quality than what is shown.
6. According to Table 3, the improvements seem limited, except for the Sampson Distance. It appears that the improved Sampson Distance results from its use in ranking. Thus, the method might not genuinely enhance overall geometric accuracy.

---

> ### Author Rebuttal · Authors · 2025-07-30
>
> We thank the reviewer for their thoughtful evaluation and recognition of our approach's intuitive appeal. We appreciate the acknowledgment that using epipolar geometry constraints for DPO is both promising and that our experiments demonstrate enhanced geometric consistency. Below we address each concern in detail:
>
> **Q1:**
> > The accuracy of the Sampson epipolar error might not be sufficient. Is it adequate for DPO? What if the two generated videos both have low epipolar accuracy?
>
> We use a combination of data filtering and a choice of training captions describing static scenes with dynamic cameras to ensure the generated videos are suitable for measuring epipolar quality. We only sample pairs for which $(\text{metric}(x_{\text{win}}) - \text{metric}(x_{\text{lose}}) > \tau) \wedge (\text{metric}(x_{\text{win}}) > \epsilon)$, which eliminates pairs where both videos have either poor ($> \epsilon$) geometry and ensures we only learn from meaningful ($> \tau$) consistency gaps. The experimental evaluation empirically validates that epipolar error provides a robust preference signal for DPO training.
>
> **Q2:**
> >Could the selection process introduce bias? For instance, some scenes might exhibit high epipolar accuracy while others show lower accuracy. This could lead to category bias after training?
>
> This is precisely why we use relative rankings rather than absolute error values. If used naively, epipolar error could introduce scene-category bias since different environments (e.g., outdoor vs. indoor). However, we only use the error for relative ranking between videos generated from the _same_ prompt (lines 210-216), ensuring no bias about scene content is inherited. The model learns geometric consistency principles rather than scene-specific error patterns
>
> **Q3:**
> > I'm curious why the training was conducted on the DL3DV and RealEstate10K datasets instead of general videos. Wouldn't this limit the model's generalization ability?
>
> We only use these datasets to create video _captions_ to prompt the model. We do not use the videos during training.
> We deliberately selected DL3DV and RealEstate10K captions because they feature dynamic cameras in static scenes, where epipolar constraints are valid. Dynamic objects would corrupt geometric measurements by violating single-camera assumptions. This is precisely one of our key insights: training on captions describing dynamic cameras in static scenes ensures high metric quality. Interestingly, the learned LoRA adapter still generalizes to diverse content, as demonstrated by improved motion stability across varied scene types, including dynamic videos, and confirmed by our evaluations.
>
> **Q4:**
> > From the attached results, I'm curious why the original video quality appears so low. Based on the reviewer's experience, Wan-2.1-T2v typically delivers much better video quality than what is shown.
>
> We used the Wan-2.1 1.3B checkpoint due to the computational cost of the 14B model. To demonstrate our model, we present examples where the generated video from the original model lacks 3D consistency, as there is little to be gained from examining videos that are already consistent.
> To better understand the base model capabilities and the impact of our alignment approach, we conducted a human evaluation. The evaluation protocol involved two steps: (1) annotators first labeled baseline videos as 3D consistent or inconsistent based on geometric artifacts and motion quality, (2) annotators then performed pairwise comparison between baseline and finetuned versions, selecting which has fewer artifacts, more stable motion, and overall quality (or "Same" if equivalent). This allows us to analyze our method's impact on both already-consistent and problematic baseline videos.
>
> **Table T1: Human Evaluation Results**
>
> | | Baseline Win | Ours Win | Same |
> |---|---|---|---|
> | **Consistent Videos** | | | |
> | Motion | 27.9% | **47.1%** | 25.0% |
> | Visual Quality | 20.6% | **47.1%** | 32.4% |
> | **Inconsistent Videos** | | | |
> | Motion | 7.5% | **60.4%** | 32.1% |
> | Visual Quality | 9.4% | **58.5%** | 32.1% |
>
> Results show our method preserves quality for already-consistent videos while dramatically improving inconsistent ones (60.4\% vs 7.5\% win rate for motion quality), confirming that our approach specifically targets geometric issues without compromising the model's existing capabilities.
>
> **Q5:**
> > What if the original methods fail to produce geometrically aligned videos? Can this method still offer improvements?}
>
> Yes. While DPO can theoretically amplify even weak geometric signals (the ranking is relative), our human evaluation shows that in practice, Wan-2.1 already often (54.1%) produces consistent videos. Our method primarily aligns the model to favor these existing coherent outputs while improving the remaining inconsistent cases, as demonstrated by the 33\% Sampson error reduction and 60.4\% human preference win rate on problematic videos.
>
> **Q6:**
> > According to Table 3, the improvements seem limited, except for the Sampson Distance. It appears that the improved Sampson Distance results from its use in ranking. Thus, the method might not genuinely enhance overall geometric accuracy.
>
> This impression arises because Table 3 has multiple goals rather than showcasing improvements: (1) it confirms we do not degrade motion quality, (2) it shows positive impact on metrics that indirectly rely on consistency (perspective fields), (3) it validates high accuracy of the Sampson metric and that we're indeed optimizing the geometric signal, and (4) it shows MET3R, a metric based on a foundational model is too noisy on generated video content. We acknowledge combining too many messages in a single table and will rework the table.

---

> > ### Comment · Area_Chair_Sz56 · 2025-08-06
> > **Author-reviewer Discussion**
> >
> > Dear reviewer,
> >
> > The system shows that you have not yet posted a discussion with the authors. As per the review guidelines, we kindly ask you to evaluate the authors’ rebuttal to determine whether your concerns have been sufficiently addressed before submitting the Mandatory Acknowledgement. If they have, please confirm with the authors. Otherwise, feel free to share any remaining questions or concerns.
> >
> > Thank you for your time and valuable feedback.
> >
> > Your AC

---

> ### Author Response · Authors · 2025-08-07
> **Following Up on Our Rebuttal Responses**
>
> Thank you again for careful review and constructive feedback. We have aimed to respond to your questions with explanations and new experimental results in our rebuttal. Having reviewed our rebuttal, are there any outstanding concerns or additional questions you would like us to clarify?

---

### Comment · Area_Chair_Sz56 · 2025-08-03
**Reviewer-Author Discussion**

Dear Reviewers,

Discussion with authors open until August 6 (AoE).

Please review the rebuttal and post any remaining concerns or comments if you have.

Kind regards,

AC

---

### Note · Authors · 2025-08-11

We sincerely thank all reviewers for their thorough evaluations, constructive feedback, and discussions throughout the review process. Your insights have significantly strengthened our work.

The key concerns raised across reviews were:
- **Baseline Model Performance**: Lack of clarity on base model capabilities and alignment impact assessment
- **Generalization**: Effectiveness on dynamic scenes, diverse content types, and real-world scenarios beyond the static training domain
- **Motion-Consistency Trade-offs**: Balance between geometric accuracy and dynamic video content
- **Metric Design Choices**: Comprehensive comparison of geometric criteria and optimization signals

Through our rebuttal, we systematically addressed each concern with extensive additional experiments and clarifications, covering all reviewer comments. We had interesting discussions with reviewers rPDn and npJ7 who maintained their positive rating. While Reviewers hTo3 and gMLo have not engaged in the discussion phase, our responses thoroughly address all questions from their original reviews:

- **Human Evaluation**: We conducted comprehensive human evaluation showing our method preserves quality on consistent videos (47.1% win rate, 25.0% same, 27.9% original model) while dramatically improving inconsistent ones (60.4% ours, 32.1% same, 7.5% original)
- **Generalization Studies**: Our extended evaluation of VideoReward and VBench benchmarks demonstrates effectiveness across diverse prompts beyond the training domain and shows generalization to dynamic videos.
- **Motion-Dynamics Analysis**: We evaluated multiple motion metrics (VBench Dynamic Degree, VideoReward Motion Quality, mean SSIM) and added static content penalty to preserve video dynamicity
- **Metric Comparisons**: We evaluated SIFT, LightGlue, SEA-Raft descriptors, and symmetric epipolar error, showing classical constraints generally provide more stable optimization signals than learned alternatives
- **3D Reconstruction Validation**: 3DGS fitting evaluation confirms geometric improvements translate to downstream applications (PSNR: 22.32→23.13, SSIM: 0.706→0.729)

Building on your valuable feedback has strengthened the paper with broader evaluations, 3D reconstruction validation, thorough methodological details, and extended geometric metric comparisons. We believe these additions address the core concerns and highlight our contribution to bridging classical geometric principles with modern video generation.

---

### Decision · Program_Chairs · 2025-09-17

**Decision:**

Reject

**Comment:**

This paper presents a method that leverages classical epipolar geometry to guide video generation, aligning with the important direction of enforcing physical inductive biases. While the rebuttal successfully addressed concerns on generalization and baselines, two critical issues remain unresolved: 1) The introduction identifies unstable motion and visual artifacts as core problems, yet the method section lacks a clear explanation of how the proposed approach specifically addresses these issues beyond imposing epipolar constraints. 2) The paper claims reducing artifacts and unstable motion as a key contribution, but provides no methodological details to substantiate how the epipolar constraints achieve this.  After careful consideration, the AC very reluctantly recommends rejection in its current form.